environmental chemistry

unsymmetrical dimethylhydrazine, degradation oxidation products, solid-phase micro-extraction, gas chromatography – mass spectrometry

**Author for correspondence:**
Xiangxuan Liu
e-mail: wdwwdw1993@163.com

This article has been edited by the Royal Society of Chemistry, including the commissioning, peer review process and editorial aspects up to the point of acceptance.

# Investigation on the compositions of unsymmetrical dimethylhydrazine treatment with different oxidants using solid-phase micro-extraction-gas chromatography – mass spectrometer

Dan Huang, Xiangxuan Liu, Xuanjun Wang, Zhiyong Huang, Zheng Xie and Huanchun Wang

High-Tech Institute of Xi'an, Shaanxi 710025, People's Republic of China

DH, 0000-0001-9302-237X

The majority of unsymmetrical dimethylhydrazine (UDMH) treatments produce lots of toxic by-products, among which *N*-Nitrosodimethylamine (NDMA) is a strong carcinogen. The compositions of the by-products are important for evaluating the treatment efficiency and understanding the UDMH degradation mechanism to achieve UDMH mineralization. The intermediate and end products of UDMH treatment with different oxidants were investigated by using a simple and fast method, solid-phase micro-extraction (SPME) in combination with gas chromatography–mass spectrometry (GC-MS). The effects of several parameters (coating fibre, salt addition, pH, sampling time and desorption time) were studied to optimize analyte recovery. The best response can be attained by the 65 $\mu$m PDMS/DVB fibre at pH 7 during 10 min after desorption of 1 min in the GC inlet. The intermediate and final oxidative products of UDMH wastewater treatment with different oxidants ($O_3$, $Mn^{2+}/O_3$, $Fe^{2+}/H_2O_2$) were investigated. The results showed that the UDMH treatment with $O_3$ could lead to high yields of NDMA. Metal catalytic ozonation could largely minimize the formation of NDMA. No NDMA was observed in the final decontaminated samples after treatment with $Fe^{2+}/H_2O_2$. The NDMA formation and degradation mechanism were discussed based on the intermediates. This study is

# 1. Introduction

Unsymmetrical dimethylhydrazine (UDMH) has been widely used as a high-energy propellant for military applications. UDMH is known as a primary eco-toxicant with the maximum permissible concentration in water as low as $0.02\,mg\,l^{-1}$ [1]. The higher health risks are associated with the rapid formation of a number of dangerous transformation products [2]. An important aspect of this problem is the wide use of different oxidants (hydrogen peroxide/copper sulfate ($Cu^{2+}/H_2O_2$) [3–5], potassium permanganate [6], chlorine reagents [7], ozone [8–11] and Fenton's reactant [12–14]) for detoxification of the wastewater containing UDMH. However, these methods produce lots of toxic by-products. Hence, much attention has been paid to the degradation products of UDMH. The reported data on the compositions of UDMH after the treatment of polluted water with different reagents are summarized in table 1. The structural formula of the products is shown in figure 1. They contain *N*-Nitrosodimethylamine (NDMA), formaldehyde dimethylhydrazone (FDMH), tetramethyltetrazene (TMT), formaldehyde methylhydrazone (FMH), dimethylamine (DMA), formic acid (HCOOH), methanol ($CH_3OH$), acetic acid ($CH_3COOH$), nitromethane (NM), *N,N*-dimethylformamide (DMF), 1,1,5-trimethylformazane (TMFN), acetaldehyde dimethylhydrazone (ADMH), and formaldehyde (HCHO). It is notable that NDMA is an ineluctable by-product of most methods, which is a strong carcinogen with a cancer risk level for drinking water $0.7\,ng\,l^{-1}$. An acceptable destruction scheme must eliminate UDMH without forming other toxic products. The efficiency of oxidants proposed for detoxification has also been estimated mainly based on the composition of UDMH degradation products. Moreover, the UDMH degradation mechanism and NDMA formation mechanism are important for NDMA controlling. The knowledge about the compositions of the intermediates and the end products of UDMH degradation is useful for NDMA formation and reduction mechanism. So, it is necessary to investigate the compounds of UDMH with different reagents.

Numerous investigations have been carried out regarding the environmental transformation effects of UDMH in water and in soil. Various analytical methods have been carried out, such as the high-performance liquid chromatography (HPLC)-MS method [15–17], gas chromatography–mass spectrometric (GC-MS) method [18–21], and solid-phase micro-extraction (SPME)-GC-MS method [22,23]. The GC-MS method provides sufficient lower detection limits for most UDMH transformation products compared to HPLC-MS methods. SPME combines sampling and sample preparation into one step. SPME is demonstrated to be rapid, simple and reproducible, with no solvent use. Moreover, SPME coupled with GC-MS provides high sensitivity. It has proven very useful for screening of UDMH transformation products in water. In addition, investigations on the intermediate and final products of UDMH are favourable to the understanding of the toxic by-product formation and degradation mechanism.

The first goal of this research was to build a multianalyte method for a simple and sensitive qualification of oxidation products in as many treated water simples as possible based on SPME in combination with GC-MS. The effects of several sampling and sample preparation parameters were studied. The method was applied in identification of the intermediate and end products of UDMH solutions before and after treatment with $O_3$, $Mn^{2+}/O_3$ and Fenton reagents. The NDMA formation and destruction mechanisms have been discussed. The results of this study are expected to provide useful data for controlling NDMA formation during UDMH wastewater treatment (figure 1).

# 2. Experimental

## 2.1. Materials

UDMH (98%, Henan Liming Chemical Reagent Co. Ltd. China), FDMH (99%, Tianjin Heowns Biochemical Co. Ltd. China), $CH_3NO_2$ (98%, Tianjin Heowns Biochemical Co. Ltd. China), TMT (99%, Tianjin Heowns Biochemical Co. Ltd. China), Dimethylamino-Acetonitrile (MSDS) (99%, Tianjin Heowns Biochemical Co. Ltd. China), 1-methyl-1,2,4-triazole (MT) (99%, Tianjin Heowns Biochemical

**Figure 1.** The structural formula of the products in table 1.

**Table 1.** Summary of UDMH degradation products via different oxidants in water.

|  | $O_3$ | $KMnO_4$ | $Cu^{2+}/H_2O_2$ | $Fe^{2+}/H_2O_2$ | chlorine reagents | cavitation |
|---|---|---|---|---|---|---|
| NDMA | + | + | + | + | + |  |
| FDMH |  |  | + |  | + |  |
| TMT |  | + | + |  | + |  |
| FMH |  |  | + |  | + |  |
| DMA |  |  |  |  | + |  |
| HCOOH |  |  |  | + |  | + |
| $CH_3OH$ | + |  | + | + | + |  |
| $CH_3COOH$ |  |  |  | + |  | + |
| NM |  |  |  | + |  | + |
| DMF |  | + | + | + | + |  |
| TMFN |  | + |  |  |  |  |
| ADMH |  |  | + |  |  |  |
| HCHO | + |  |  | + |  |  |

Co. Ltd. China), DMF (99%, Tianjin Heowns Biochemical Co. Ltd. China), $CH_3OH$ (99%, Tianjin Heowns Biochemical Co. Ltd. China), $CH_3COOH$ (88%, Tianjin Heowns Biochemical Co. Ltd. China), HCHO (37.0%−40.0%, Tianjin Heowns Biochemical Co. Ltd. China), DMA (2.0 M in Methanol, Shanghai Mackin Chemical Co. Ltd. China), NDMA (1000 $\mu$g l$^{-1}$ in Methanol, Accustandard Co. Ltd. USA), NaCl (Analytical grade, Sinopharm Chemical Co. Ltd. China), $Fe_2SO_4 \cdot 7H_2O$ (analytical grade, Sinopharm Chemic al Co. Ltd. China) and $H_2O_2$ (30%, Sinopharm Chemical Co. Ltd. China) were commercially available. Ozone was generated by a laboratory ozonizer with an oxygen source (ModelCFK-3, Hangzhou Rongxin Electronic Equipment Co. Ltd. China).

In all of the experiments, deionized water was used to prepare the solutions. Measurement of the pH was done by a pH meter (PHSJ-4A, Shanghai Yidian Scientific Instrument Co. Ltd. China). The degradation samples were treated with an SPME device (85 $\mu$m SPME needle, USA) and were analysed by a GC/MS (Agilent 7890A-5795C, USA).

## 2.2. Reaction procedure

UDMH ozonation procedure: 200 ml of UDMH (1 g l$^{-1}$) was added to the reaction chamber. Then, the reaction chamber was flushed with ozone (the production of ozonizer, 3 g h$^{-1}$).

UDMH ozonation with catalyst $Mn^{2+}$ procedure: 200 ml of UDMH (1 g l$^{-1}$) and 0.755 g of $MnSO_4$ were added to the reaction chamber in order. Then, the reaction chamber was flushed with ozone.

UDMH oxidation by Fenton reagents ($Fe^{2+}/H_2O_2$) procedure: 100 ml of UDMH (1 gl$^{-1}$), 4.08 ml of 30% $H_2O_2$, 7.8 ml of $FeSO_4 \cdot 7H_2O$ (0.5 mol l$^{-1}$) were added to the reaction chamber in that order. An appropriate amount of distilled water was added to make a volume of 200 ml, so that the

concentration of dimethyl hydrazine in the flask was $1000\,\mathrm{mg\,l^{-1}}$. At 20 min, 50 ml of the reaction solution was filtered and analysed. And the reaction mixture was added with 30% $H_2O_2$ (3.06 ml) and $FeSO_4 \cdot 7H_2O$ (5.85 ml).

Each experiment was repeated three to five times.

## 2.3. Samples

Fresh samples were prepared by mixing 50.0 ml of water with 100 µl of a standard solution of analytes in a 100 ml vial followed by sealing of the vial and agitation for 30 min. The fresh samples contained 12 oxidative products of UDMH.

The reaction samples were filtered at different times. All SPME fibres were conditioned in the GC inlet at 200°C before use. SPME-based sample desorption was carried out at 200°C for 30 min in the splitless mode. After adjusting the pH of the solution following the reaction, 8 ml of the sample was placed into a 10 ml beaker, and sodium chloride (NaCl) was added. The small beaker was placed on a magnetic stirrer. After the sodium chloride was completely dissolved, the SPME fibre was immersed in the solution with different times for adsorption equilibrium. Finally, the SPME fibre was inserted into the GC inlet by the SPME syringe.

## 2.4. Parameters of GC-MS analyses

The optimal injector temperature was 200°C. The separation was performed on a DB-225S (30 m × 250 µm × 0.25 µm) capillary column. The column was maintained at a constant flow of UHP-grade helium at $1.5\,\mathrm{ml\,min^{-1}}$. The GC oven programme was as follows: 35°C for the first 5 min, followed by $10°\mathrm{C\,min^{-1}}$ ramping to 15°C. The split ratio was 1 : 1.

The MS was operated in the full scan mode from 19 to 550 $m/z$. The ion source was held at 230°C and the electron multiplier was set at 70 eV.

## 2.5. Statistical analysis

The peak area was calculated by an RET integrator. The minimum integral is 0.1 of the maximum area, which is used as the integral parameter to ensure that the peak with a smaller area can be effectively integrated. Repeatability of the applied method was confirmed by the analysis of selected samples in triplicate. RSDs for replicates did not exceed 15%.

# 3. Results and discussion

## 3.1. Optimization of the sampling condition

The objective of this chapter is to screen UDMH in as many degradation oxidative products as possible and to obtain the best responses based on SPME-GC/MS. The effects of several sampling and sample parameters were studied. These involved SPME fibre coating type, effects of added salt, SPME sampling pH and time and optimization of GC injection/desorption.

### 3.1.1. Selection of SPME fibre coating

Polymeric coatings used in SPME vary in their layer materials and thicknesses, which affect SPME sensitivity towards specific chemical groups of compounds. The main coating materials, which are polydimethylsilane (PDMS), divinylbenzene (DVB), polyethylene glycol (CW), polyacrylate (PA) and activated carbon (Carboxen, CAR), are currently widely used. The most suitable SPME fibre coating for sampling of UDMH and its degradation products from water were selected from 65 µm PDMS/DVB, 85 µm CAR/ PDMS, 85 µm PA and 100 µm PDMS tested for identifying the main UDMH degradation oxidative products. Extractions with SPME were carried out from the samples at 30 min. As shown in figure 2, the best response and range of extracted chemicals were obtained by applying the 65 µm PDMS/DVB.

There were 12 kinds of UDMH and their oxidative products (DMA, FDMH, $CH_3OH$, UDMH, NM, TMT, $CH_3COOH$, MSDS, NDMA, DMF, MT, HCHO) in the fresh sample. Except for HCHO, they could all be detected. The identified transformation products are listed in table 2. Therefore, a 65 µm PDMS/ DVB fibre was selected for further studies. Because the peak shapes of DMA, $CH_3OH$ and $CH_3COOH$ were unstable and poor, others were selected for further studies.

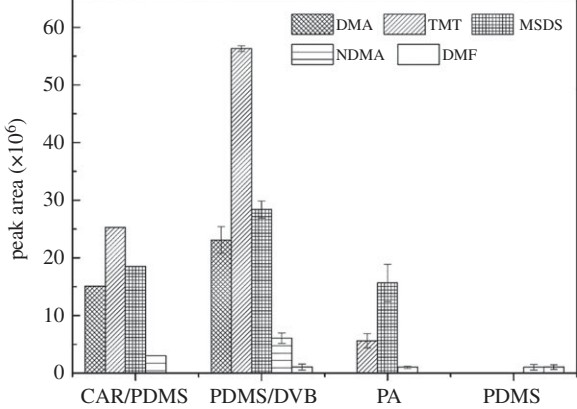

**Figure 2.** The response to the fresh sample with four different SPME fibres.

**Table 2.** The compositions detected by SPME-GC/MS.

| product | retention time | CAS no | Pro. (%) |
| --- | --- | --- | --- |
| DMA | 1.053 | 124-40-3 | 78.5 |
| FDMH | 2.304 | 2035-89-4 | 80 |
| $CH_3OH$ | 2.731 | 67-56-1 | 87.5 |
| UDMH | 3.643 | 57-14-7 | 96.7 |
| NM | 4.429 | 75-52-5 | 97.8 |
| TMT | 6.610 | 6130-87-6 | 70.5 |
| $CH_3COOH$ | 6.813 | 2035-89-4 | 91.4 |
| MSDS | 7.830 | 924-64-7 | 97.4 |
| NDMA | 9.430 | 62-75-9 | 90.8 |
| DMF | 9.945 | 68-12-2 | 80.8 |
| MT | 11.503 | 6086-21-1 | 88 |

### 3.1.2. Effects of salt addition

The reports [2,23] showed that the water content in soil plays a significant role in the recovery of all UDMH transformation products and makes the quantification process difficult or even impossible. Salt was used to improve sample recovery, because the addition of salt could enhance the amount extracted by the fibre [24]. To investigate the effects of salt on the extraction efficiency of UDMH and its oxidation products, the different salt (NaCl) concentrations (0%, 10%, 20%, 30%, 40%) were selected to add to the samples. The results are shown in figure 3.

The amount of salt added resulted in increased recovery of almost all analytes. The increased recovery was observed when salt concentration increased from 10% to 30% while at a salt concentration of 40% (an oversaturation solution of NaCl) the recovery decreased. It was obvious that the peak recovery occurred with the salt concentration of NaCl (30%). The salt concentration of NaCl (30%) was therefore chosen to prepare the samples.

### 3.1.3. Effects of pH

pH plays a significant role in compound distribution between cation and neutral molecule, which affects the extraction effect. So the effect of pH on the peak area of degradation products was studied at 5.0, 7.0 and 10.0. The results are presented in figure 4.

As we can see from figure 4, TMT could not be observed and the response of UDMH was poor at pH 5.0. The recovery of UDMH and its oxidative products at pH 7.0 was similar to that at pH 10.0. Generally,

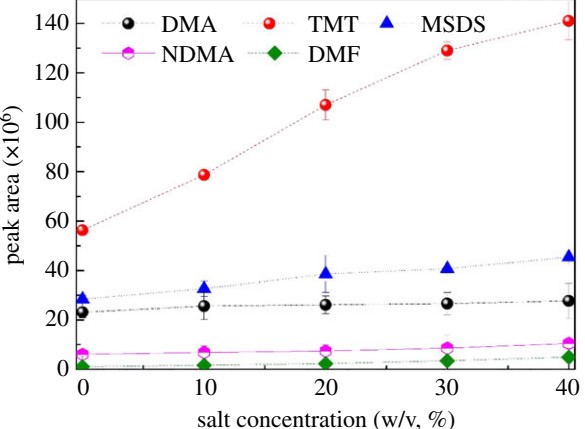

**Figure 3.** Effect of salt concentration on extraction.

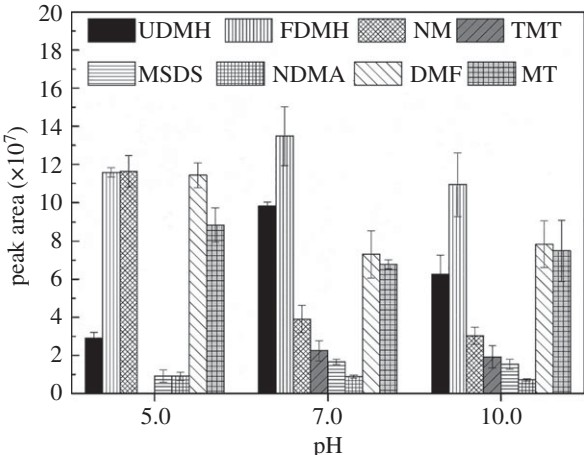

**Figure 4.** Effect of pH on extraction efficiency.

the solution is under neutral condition at pH 7.0. Thus, SPME at pH 7.0 provides the best combination of convenience and sensitivity.

### 3.1.4. Effects of SPME sampling time

SPME is often used for equilibrium sampling, resulting in an equilibrium between the analytes in the aqueous phase and the fibre, and also including any other matrix capable of binding the analyte. The extraction time was determined based on sufficient response and reproducibility within a practical time frame. Several sampling times (1 min, 5 min, 10 min, 20 min and 30 min) were studied to determine the practical SPME sampling time. Mass detector responses versus SPME sampling time are presented in figure 5. The increase of peak area with the increase of sampling time was observed for all the analytes. No significant difference in the area counts was found ranging from 10 to 30 min. The extraction time was, therefore, chosen to be 10 min which resulted in sufficient signal and precision for all the detected UDMH oxidation products.

### 3.1.5. Optimization of SPME desorption time

Studies have shown that excessive desorption time will cause the fibre coating to degrade slowly in a high-temperature environment, affecting the service life of the extracted fibre. But, too short a desorption time cannot fully release the target component, resulting in reduced desorption efficiency and material residue.

In this experiment, different desorption times (10 s, 30 s, 1 min, 2 min and 5 min) were selected for the experiment. The results are shown in figure 6. The extension of desorption time has little effect on the mass detector responses. Moreover, after the secondary desorption of the extracted fibres, it was found that the six

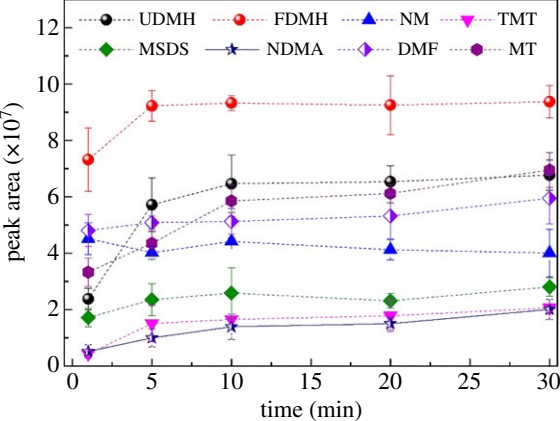

**Figure 5.** Effect of SPME sampling time on extraction.

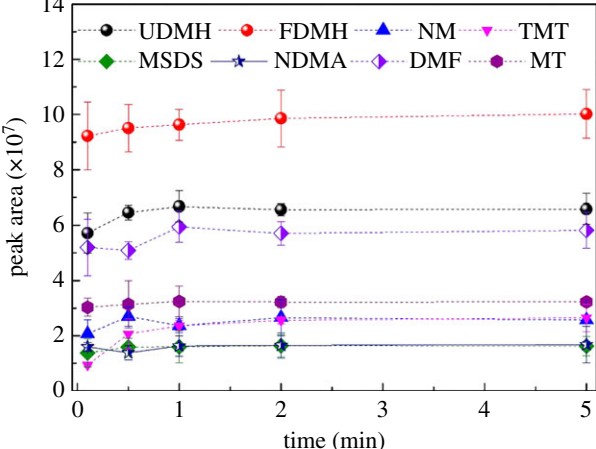

**Figure 6.** Effect of SPME desorption time on extraction.

substances did not re-peak again, which indicated that when the GC inlet temperature was set to 250°C, the eight kinds of odours adsorbed on the fibres were obtained using the desorption time of 1 min. The substance is completely desorbed. Therefore, the optimal SPME desorption time is 1 min.

## 3.2. MDL of this method

Generally, the revised definition of the method detection limit (MDL) [25] is an estimate of the measured concentration on which there is 99% confidence that a given analyte is present in the given sample matrix. Here, the MDL is roughly determined by

$$\mathrm{MDL} = k_i s_i \frac{c}{\bar{X}}, \tag{3.1}$$

where $k_i$ is the confidence factor, generally 3, $s_i$ is the standard deviation of the sample measurement reading. $c$ is a sample content value and $\bar{X}$ is the average value of sample measurement readings.

As shown in table 3, SPME-GC/MS allowed the MDL of UDMH, NM, DMF and MT to be as low as 9.3 mg l$^{-1}$, 3.3 mg l$^{-1}$, 5.0 mg l$^{-1}$ and 7.1 mg l$^{-1}$ respectively. The MDL of FDMH and NDMA is as low as 20 µg l$^{-1}$. The MDL of TMT as low as 6.9 µg l$^{-1}$.

## 3.3. Identification of the products of the treated water

Here, we selected three oxidative reagents ($O_3$, $Mn^{2+}/O_3$, $Fe^{2+}/H_2O_2$) to treat UDMH wastewater. The compositions of the fresh UDMH solutions and solutions after treatment with oxidative reagents were analysed using GC-MS based on SPME.

**Table 3.** MDL of UDMH and its degradation products.

| compound | peak area | | average value | $S_i$ | MDL (mg l$^{-1}$) |
|---|---|---|---|---|---|
| UDMH | 11083274 | 10562878 | 11065104.5 | 842288.8 | 9.3 |
|  | 11843866 | 10025687 |  |  |  |
|  | 11568974 | 10025698 |  |  |  |
|  | 12365897 | 11044562 |  |  |  |
| FDMH | 14435654 | 15579654 | 15150937.4 | 1095812.3 | 0.021 |
|  | 16129543 | 15388590 |  |  |  |
|  | 15532927 | 13140917 |  |  |  |
|  | 14435654 | 16564560 |  |  |  |
| TMT | 106618338 | 139934000 | 117694271.3 | 13520422.8 | 0.0069 |
|  | 122036490 | 130536890 |  |  |  |
|  | 104620609 | 100700160 |  |  |  |
|  | 121498760 | 115608923 |  |  |  |
| NM | 7001567 | 7750996 | 7308419.75 | 401052.7 | 3.3 |
|  | 7713547 | 6464465 |  |  |  |
|  | 7128949 | 7402588 |  |  |  |
|  | 7458925 | 7546321 |  |  |  |
| NDMA | 6498546 | 7386548 | 6983683.75 | 257010.1 | 0.011 |
|  | 6895268 | 7025864 |  |  |  |
|  | 6985474 | 6854956 |  |  |  |
|  | 7056982 | 7165832 |  |  |  |
| DMF | 2711471 | 2904413 | 2938829 | 245401.7 | 5.0 |
|  | 2714194 | 3270874 |  |  |  |
|  | 3349445 | 2746290 |  |  |  |
|  | 2918300 | 2895645 |  |  |  |
| MT | 1872495 | 1974081 | 2158139.8 | 254836.0 | 7.1 |
|  | 1927596 | 2342147 |  |  |  |
|  | 2435296 | 2557846 |  |  |  |
|  | 2114088 | 2041569 |  |  |  |

### 3.3.1. $O_3$

Figure 7 shows the composition of extracts from fresh UDMH solutions with a concentration of 1 g l$^{-1}$ after being treated with ozone.

It follows from figure 7 that there was no UDMH in the extract after purification within 20 min. The extract also contained DMA, FDMH, TMT, MSDS, NDMA and DMF. After treatment with 6 h, the main product was NDMA, which was higher than the samples at 20 min. Others are DMA, FDMH, DMF and NM. It was discovered that after treatment with ozone, the solution contained no UDMH but did contain high yields of NDMA, which poses a high risk for human beings. This implies that $O_3$ plays an important role in NDMA formation from UDMH.

The NDMA formation mechanism during wastewater ozonation has attracted much attention [10,26–28]. Transformation of UDMH to NDMA is mainly induced by ozone or HO· rather than the dissolved oxygen proposed previously. Pisarenko *et al.* [28] found that wastewater with higher HO· scavenging may also lead to higher NDMA formation during ozonation. Radiolysis experiments [28] were performed to isolate the contribution of hydroxy radicals. During radiolysis experiments, there was no NDMA formation observed in any of the wastewater samples. The results of these experiments confirm that NDMA formation is due to reactions associated with molecular

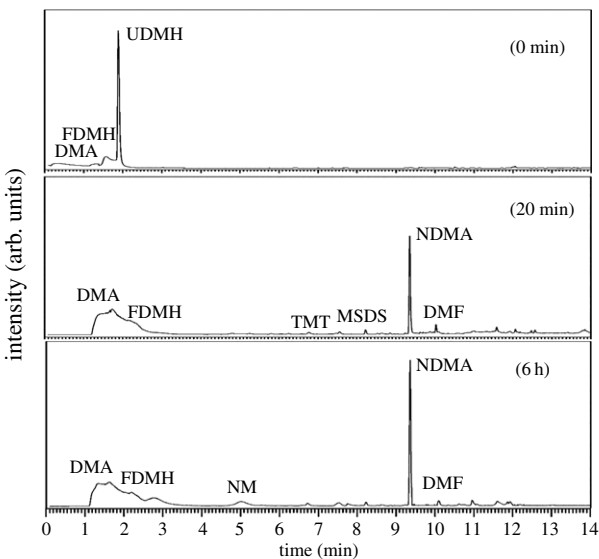

**Figure 7.** Gas chromatogram of an extraction of a fresh UDMH solution (1 g l$^{-1}$) treated with O$_3$ at different times.

ozone. Our results are consistent with these results. Hence, the NDMA formation mechanism was proposed by the following:

$$H_3C\diagdown N-N\diagup H + O_3 \longrightarrow H_3C\diagdown N-N^{\cdot} + HO^{\cdot} + O_2 \qquad (3.2)$$

$$H_3C\diagdown N-N\diagup H + O_3 \longrightarrow H_3C\diagdown N-N^+\diagup O^- H + O_2 \qquad (3.3)$$

$$H_3C\diagdown N-N^+\diagup O^- H \xrightarrow{O_3} H_3C\diagdown N-N=O \qquad (3.4)$$

$$H_3C\diagdown N-N^{\cdot}\diagup H \xrightarrow{O_3} H_3C\diagdown N-N=O \qquad (3.5)$$

The reaction of ozone with UDMH is thought to proceed via H-abstraction from $-NH_2$ or via O-addition to $-NH_2$ and the UDMH intermediate or UDMH radical form. These intermediates could be through further H-atom abstraction or O-atom addition and finally form NDMA.

### 3.3.2. Mn$^{2+}$/O$_3$

Catalytic ozonation has recently gained significant attention as an effective process used for organic removal from water [29,30]. Among the types of metal catalysis ozonation that have been applied to homogeneous catalysis, Mn$^{2+}$ has gained significant attention because of its excellent catalytic performance, with many studies focused on the degradation effects of organic pollutants [31,32]. This research used Mn$^{2+}$ as a catalyst and investigated the oxidative products of UDMH after treatment by the Mn$^{2+}$/O$_3$ system. The results are shown in figure 8.

When the UDMH was treated with Mn$^{2+}$/O$_3$ after 20 min, the UDMH was almost consumed and significant amounts of TMT formed. When the UDMH was treated after 6 h, the significant amounts of TMT were almost consumed. The solutions still contained NDMA and DMA.

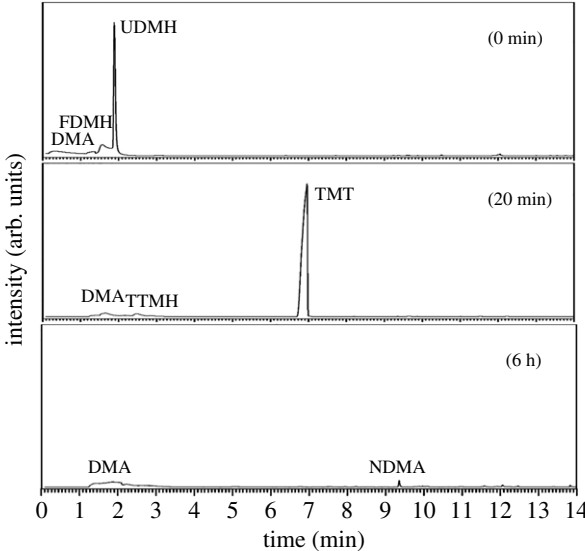

**Figure 8.** Gas chromatogram of an extraction of a fresh UDMH solution (1 g l$^{-1}$) before and after treatment with Mn$^{2+}$/O$_3$ at different times.

The oxidative degradation products are different from those of UDMH ozonation. In addition, the yields of NDMA in the presence of Mn$^{2+}$ are much lower than those in the absence of Mn$^{2+}$. Except for direct ozonation, the mechanism of Mn$^{2+}$ catalytic ozonation of UDMH includes oxidation by hydroxyl radical produced from the decomposition of dissolved ozone, and oxidation by Mn$^{4+}$ and Mn$^{7+}$. It was also found that the hydroxyl radicals were the main reactant in neutral or alkaline conditions.

In this article, the UDMH wastewater is alkaline. So the hydroxyl radicals play a key role in UDMH destruction. From the composition of the products, TMT is the important intermediate. As we know, NDMA formation is generally initialled by oxidation of −NH$_2$ group. So the TMT formation pathway was proposed as follows:

$$\ce{(H3C)2N-NH2 + HO^{\bullet} -> (H3C)2N-NH^{\bullet} + H2O} \tag{3.6}$$

$$\ce{(H3C)2N-NH^{\bullet} ->[HO^{\bullet}] (H3C)2N^{+}=N^{-} -> (H3C)2N-N=N-N(CH3)2} \tag{3.7}$$

HO is the main abstract H-atom of the −NH$_2$ group. This is the reason for the large amounts of TMT formation within 20 min. Koji [9] *et al.* reported that the NDMA formation yield of TMT during ozonation was 19%. The end products of NDMA formation were proposed to be through the intermediate TMT oxidation.

$$\ce{(H3C)2N-NH2 + HO^{\bullet} -> (H2C^{\bullet})(H3C)N-NH2 + H2O} \tag{3.8}$$

$$\ce{(H3C)2N-N=N-N(CH3)2 ->[O3][HO^{\bullet}] (H3C)2N-N=O} \tag{3.9}$$

### 3.3.3. Fe$^{2+}$/H$_2$O$_2$

The possibility of decontaminating UDMH with Fe$^{2+}$/H$_2$O$_2$ has been studied. The oxidation products are shown in figure 9. A variety of intermediate products can be observed in the samples after UDMH treatment for 20 min. DMA, TMT, MSDS, NDMA and DMF were observed. Then, the experimental

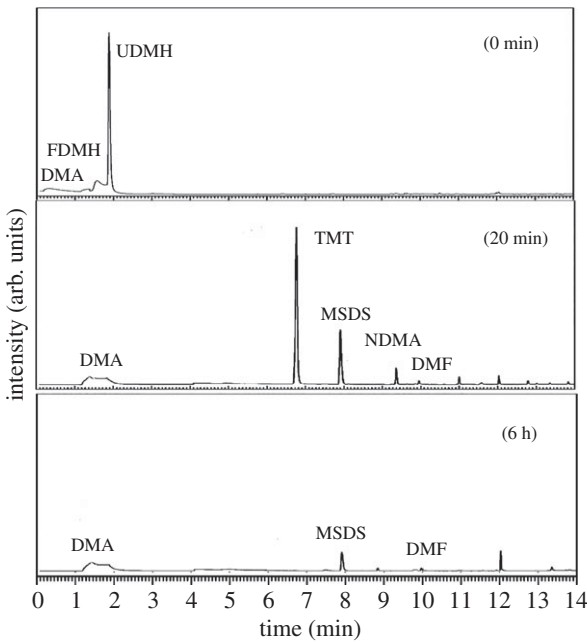

**Figure 9.** Gas chromatogram of an extraction of a fresh UDMH solution (1 g l$^{-1}$) treated with Fe$^{2+}$/H$_2$O$_2$ at different times.

samples were again added with the Fenton reagents. The end products of UDMH were DMA, MSDS and DMG after treatment with Fenton reagents for 6 h.

Previous study results showed that there was no NDMA formation observed during decontamination of UDMH by heterogeneous Fenton system [12], cavitation [33] or cavitation-induced advanced Fenton process [14]. In our experimental results, NDMA was detected in the intermediate products. This indicates that NDMA does form without ozone. UDMH radical can further H-abstraction or O-addition to form NDMA by HO·

$$\text{(3.10)}$$

However, there was no NDMA formation observed in the end products. This indicates that the degradation of NDMA can be a result of the HO·. We suppose that it may be initialled by H-abstraction from the $-CH_3$ group according to the following reactions. Previous studies have shown that NDMA destruction by Fenton's reagent is most efficient at low pH. Minakata [34] *et al.* proposed that NDMA has three potential initial degradation mechanisms: (i) H atom abstraction from a C−H bond of the methyl group, (ii) HO· addition to amine nitrogen, and (iii) HO· addition to nitrosyl nitrogen.

$$\text{(3.11)}$$

$$\text{(3.12)}$$

$$\text{(3.13)}$$

According to their quantum mechanical calculation results, H-abstraction from C−H bond of the methyl functional group of NDMA is the dominant initial reaction pathway as induced by HO. So it is necessary for the treatment method to add the Fenton reagents to the intermediate products. NDMA can be totally destructed.

The results are compared on identification of UDMH degradation products in wastewater with the previously published data [6,12,14,33]. The use of SPME-GC/MS results in the detection of a total of 11 compounds, which is more than previously reported. This method could target different

compounds with one sample. It allowed us to obtain more comprehensive data on identification of UDMH degradation products. This information about the degradation products is useful for understanding the mechanism. Quantitative determination of UDMH degradation products using this method needs to be studied further.

# 4. Conclusion

The main intermediate and end products of UDMH after treatment were qualitatively analysed by a new sample and fast method, SPME-GC/MS. The effects of several sampling and sample preparation parameters were studied. These involved SPME fibre type, salt addition, pH, sampling time and desorption time. It was determined that the 65 µm PDMS/DVB SPME fibre coating provides the highest selectivity for detection of UDMH oxidative products and provides the best combination of convenience and sensitivity achieved at pH 7, sampling time 10 min and desorption time 1 min.

The intermediate and end products of UDMH after treatment with $O_3$, $Mn^{2+}/O_3$, $Fe^{2+}/H_2O_2$ have been investigated. The results showed that the treatment with $O_3$ produces large amounts of NDMA. Metal catalytic ozonation ($Mn^{2+}/O_3$) could largely minimize the formation of NDMA. This is mainly due to the fact that the system generates oxidative free radicals (hydroxyl radicals), which plays a key role in H-atom abstraction from the $-NH_2$ group of UDMH but not in O-addition, finally leading to high yields of TMT. The NDMA in the end products may be formed from TMT oxidation. The intermediate products after treatment with $Fe^{2+}/H_2O_2$ contain NDMA. NDMA can be destructed by further oxidation via HO. The ozone plays a key role in NDMA formation, while the HO· makes a significant contribution to NDMA destruction.

Data accessibility. This article does not contain any additional data.

Authors' contributions. D.H. carried out the molecular laboratory work, participated in data analysis, carried out sequence alignments, participated in the design of the study and drafted the manuscript; X.L. carried out the statistical analyses and critically revised the manuscript; Z.X. critically revised the manuscript; X.W., Z.H. and H.W. conceived of the study, designed the study, coordinated the study and helped draft the manuscript. All authors gave final approval for publication and agreed to be held accountable for the work performed therein.

Competing interests. The authors declare no conflict of interest.

Funding. This study was funded by China Postdoctoral Science Foundation (grant no. 2016M600084).

Acknowledgement. The authors are highly grateful to Mr Wang for the kind guidance.

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
