## [Reviewer comments · Royal Society Open Science]

Review History

RSOS-190263.R0 (Original submission)

Review form: Reviewer 1

Is the manuscript scientifically sound in its present form?

No

Are the interpretations and conclusions justified by the results?

No

Is the language acceptable?

No

Is it clear how to access all supporting data?

Yes

Do you have any ethical concerns with this paper?

No

Have you any concerns about statistical analyses in this paper?

I do not feel qualified to assess the statistics

Recommendation?

Major revision is needed (please make suggestions in comments)

Comments to the Author(s)

Manuscript entitled "Investigation on the Compositions of Unsymmetrical Dimethylhydrazine Treatment with different Oxidants Using SPME-GC/MS" determines the intermediates and oxidation products of Unsymmetrical Dimethylhydrazine (UDMH) reacted with different oxidants (O₃, Mn²⁺/O₃, Fe²⁺/H₂O₂) and optimizes parameters involved coating fiber, salt addition, pH, sampling time and desorption time. The UDMH reacted with O₃ leads to high yields of N-Nitrosodimethylamine (NDMA). Metal catalytic ozonation can largely minimize the formation of NDMA. After treatment with Fe²⁺/H₂O₂, the final products of UDMH have no NDMA. The NDMA formation and degradation mechanism were discussed based on the intermediates. This study is expected to provide useful information for controlling NDMA formation during UDMH wastewater treatment.

Overall, the investigation is original. The paper is not well-written and has many grammar and formatting problems. As a result, I think the current manuscript is not suitable for the Journal of Royal Society Open Science, there are some comments should be considered:

Specific comments:

1. The highlight of this paper is the use of SPME-GC/MS, but the description of it in the introduction is too simple. Could the authors supplement some advantages of using SPME-GC/MS in the introduction part?
2. Page 3, table 1 gives some abbreviation of UDMH degradation products. Could the authors supplement the structural formula of products in table 1?
3. Page 4, line 28, the reaction chamber was flushed with ozone (3 g/h). How to determine the concentration (3 g/h) of ozone dissolved in water during ozonation?
4. Page 6, line 13, the author directly proposed the best choice was 65 μ m PDMS / DVB instead of 85 μ m CAR/PDMS, 85 μ m PA, and 100 μ m PDMS. Could the authors provide some data which can explain that 65 μ m PDMS/DVB is the best choice?
5. Page 6, line 48, the authors mentioned: "Addition of salt was used to compensate on sample recovery". Please give more explanation of why adding salt can increase recovery.
6. Page 12, in the Fig.5, the peaks of DMA and FDMH are not separated, is this related to the temperature rise procedure of GC/MS?
7. Page 12, line 60, was the radiolysis experiments done by yourself or by reference? Please indicate if it comes from reference.
8. Page 13, line 6, the authors mentioned "These experiments results confirm that NDMA formation is due to reactions associated with molecular ozone. Our results are inconsistent with these results." I think this statement is confused.

Small comments:

1. There should be a space between the number and the unit.
2. Introduction and conclusion parts of the text should be aligned at both ends.
3. The author should carefully check the content and formatting problems.
4. Page 1, line 19, "N-nitrosodimethylamine" should be "N-Nitrosodimethylamine".

Review form: Reviewer 2

Is the manuscript scientifically sound in its present form?

No

Are the interpretations and conclusions justified by the results?

Yes

Is the language acceptable?

No

Is it clear how to access all supporting data?

Not Applicable

Do you have any ethical concerns with this paper?

No

Have you any concerns about statistical analyses in this paper?

Yes

Recommendation?

Major revision is needed (please make suggestions in comments)

Comments to the Author(s)

This manuscript describes investigation on the compositions of unsymmetrical dimethylhydrazine treatment with different oxidants using SPME-GC/MS. The manuscript needs major revision. The manuscript requires many edits, carefully proof reading, improvement in English.

Throughout the whole manuscript one space should be given between number and units, e.g. Page 1, line 33, 65 μ m should be 65 μ m; Page 2, line 14, 0.02mg/L[1] should be 0.02 mg/L [1]. There should one space before the brackets.

Line 28-32, Formic acid(HCOOH), Methanol(CH₃OH), Acetic acid(CH₃COOH), Nitromethane (NM), N,N-dimethylformamide (DMF), 1,1,5-trimethylformazane (TMFN), Acetaldehyde dimethylhydrazone (ADMH), Formaldehyde (HCHO).

The compound names should be in lower case letters.

Page 4, line 52, Modern samples were samples were... Authors should define what they mean by Modern sample? Is it fresh sample, if so use fresh sample...

Page 5, line 30, before "Results and Discussion" a new sub-section "Statistical analysis" should be include to describe the statistical treatment of data.

Page 6, line 7, The modern sample contains....???

Page 6, line 44, There were reports...give [Ref. -- ??]

Page 7, Fig. 1, x-axis label should be corrected as Salt concentration (w/v, %)

Page 10, line 46, 9.340mg/L, 3.292mg/L, 5.010mg/L and 7.084mg/L...MDL data should be given in two significant figures only, e.g. 9.3 mg/L, 3.3 mg/L, 5.0 mg/L and 7.1 mg/L...

Page 11, Table 3, column 5, MDL should be given in two significant figures only.

The data in column 2-4 should be given in three significant figures only.

Page 17, line 27, before Conclusion, the advantages and disadvantages of the proposed method should be given in comparison to the literature reported methods.

Decision letter (RSOS-190263.R0)

12-Mar-2019

Dear Dr Huang:

Title: Investigation on the Compositions of Unsymmetrical Dimethylhydrazine Treatment with Different Oxidants Using SPME-GC/MS
Manuscript ID: RSOS-190263

The editor assigned to your manuscript has now received comments from reviewers. We would like you to revise your paper in accordance with the referee and Subject Editor suggestions which can be found below (not including confidential reports to the Editor). Please note this decision does not guarantee eventual acceptance.

Please submit your revised paper before 04-Apr-2019. Please note that the revision deadline will expire at 00.00am on this date. If we do not hear from you within this time then it will be assumed that the paper has been withdrawn. In exceptional circumstances, extensions may be possible if agreed with the Editorial Office in advance. We do not allow multiple rounds of revision so we urge you to make every effort to fully address all of the comments at this stage. If deemed necessary by the Editors, your manuscript will be sent back to one or more of the original reviewers for assessment. If the original reviewers are not available we may invite new reviewers.

RSC Associate Editor:
Comments to the Author:
(There are no comments.)

RSC Subject Editor:
Comments to the Author:
(There are no comments.)

Reviewers' Comments to Author:
Reviewer: 1

Comments to the Author(s)

Manuscript entitled "Investigation on the Compositions of Unsymmetrical Dimethylhydrazine Treatment with different Oxidants Using SPME-GC/MS" determines the intermediates and oxidation products of Unsymmetrical Dimethylhydrazine (UDMH) reacted with different oxidants (O₃, Mn²⁺/O₃, Fe²⁺/H₂O₂) and optimizes parameters involved coating fiber, salt addition, pH, sampling time and desorption time. The UDMH reacted with O₃ leads to high yields of N-Nitrosodimethylamine (NDMA). Metal catalytic ozonation can largely minimize the formation of NDMA. After treatment with Fe²⁺/H₂O₂, the final products of UDMH have no NDMA. The NDMA formation and degradation mechanism were discussed based on the intermediates. This study is expected to provide useful information for controlling NDMA formation during UDMH wastewater treatment.

Overall, the investigation is original. The paper is not well-written and has many grammar and formatting problems. As a result, I think the current manuscript is not suitable for the Journal of Royal Society Open Science, there are some comments should be considered:

Specific comments:

1. The highlight of this paper is the use of SPME-GC/MS, but the description of it in the introduction is too simple. Could the authors supplement some advantages of using SPME-GC/MS in the introduction part?
2. Page 3, table 1 gives some abbreviation of UDMH degradation products. Could the authors supplement the structural formula of products in table 1?
3. Page 4, line 28, the reaction chamber was flushed with ozone (3 g/h). How to determine the concentration (3 g/h) of ozone dissolved in water during ozonation?
4. Page 6, line 13, the author directly proposed the best choice was 65 μm PDMS / DVB instead of 85 μm CAR/PDMS, 85 μm PA, and 100 μm PDMS. Could the authors provide some data which can explain that 65 μm PDMS/DVB is the best choice?
5. Page 6, line 48, the authors mentioned: "Addition of salt was used to compensate on sample recovery". Please give more explanation of why adding salt can increase recovery.
6. Page 12, in the Fig.5, the peaks of DMA and FDMH are not separated, is this related to the temperature rise procedure of GC/MS?
7. Page 12, line 60, was the radiolysis experiments done by yourself or by reference? Please indicate if it comes from reference.

8. Page 13, line 6, the authors mentioned "These experiments results confirm that NDMA formation is due to reactions associated with molecular ozone. Our results are inconsistent with these results." I think this statement is confused.

Small comments:

1. There should be a space between the number and the unit.
2. Introduction and conclusion parts of the text should be aligned at both ends.
3. The author should carefully check the content and formatting problems.
4. Page 1, line 19, "N-nitrosodimethylamine" should be "N-Nitrosodimethylamine".

Reviewer: 2

Comments to the Author(s)

This manuscript describes investigation on the compositions of unsymmetrical dimethylhydrazine treatment with different oxidants using SPME-GC/MS. The manuscript needs major revision. The manuscript requires many edits, carefully proof reading, improvement in English.

Throughout the whole manuscript one space should be given between number and units, e.g. Page 1, line 33, 65 μ m should be 65 μ m; Page 2, line 14, 0.02mg/L[1] should be 0.02 mg/L [1]. There should one space before the brackets.

Line 28-32, Formic acid(HCOOH), Methanol(CH₃OH), Acetic acid(CH₃COOH), Nitromethane (NM), N,N-dimethylformamide (DMF), 1,1,5-trimethylformazane (TMFN), Acetaldehyde dimethylhydrazone (ADMH), Formaldehyde (HCHO).

The compound names should be in lower case letters.

Page 4, line 52, Modern samples were samples were... Authors should define what they mean by Modern sample? Is it fresh sample, if so use fresh sample...

Page 5, line 30, before "Results and Discussion" a new sub-section "Statistical analysis" should be include to describe the statistical treatment of data.

Page 6, line 7, The modern sample contains....???

Page 6, line 44, There were reports...give [Ref. -- ??]

Page 7, Fig. 1, x-axis label should be corrected as Salt concentration (w/v, %)

Page 10, line 46, 9.340mg/L, 3.292mg/L, 5.010mg/L and 7.084mg/L...MDL data should be given in two significant figures only, e.g. 9.3 mg/L, 3.3 mg/L, 5.0 mg/L and 7.1 mg/L...

Page 11, Table 3, column 5, MDL should be given in two significant figures only.

The data in column 2-4 should be given in three significant figures only.

Page 17, line 27, before Conclusion, the advantages and disadvantages of the proposed method should be given in comparison to the literature reported methods.

Author's Response to Decision Letter for (RSOS-190263.R0)

See Appendices A & B.

Decision letter (RSOS-190263.R1)

26-Mar-2019

Dear Dr Huang:

Title: Investigation on the Compositions of Unsymmetrical Dimethylhydrazine Treatment with Different Oxidants Using SPME-GC/MS

Manuscript ID: RSOS-190263.R1

It is a pleasure to accept your manuscript in its current form for publication in Royal Society Open Science. The chemistry content of Royal Society Open Science is published in collaboration with the Royal Society of Chemistry.

RSC Associate Editor
Comments to the Author:
(There are no comments.)

Reviewer(s)' Comments to Author:

Appendix A

Dear reviewer,

I am appreciate for your comments and suggestions. The manuscript has been revised based on your comments. The responses of your questions were as follows:

1. The highlight of this paper is the use of SPME-GC/MS, but the description of it in the introduction is too simple. Could the authors supplement some advantages of using SPME-GC/MS in the introduction part?

R: The advantages have been given in the introduction, which are marked green in page 3 line 4,5.

2. Page 3, table 1 gives some abbreviation of UDMH degradation products. Could the authors supplement the structural formula of products in table 1?

R: The structural formula of products in table1 have been added as shown in Fig.1.

3. Page 4, line 28, the reaction chamber was flushed with ozone (3 g/h). How to determine the concentration (3 g/h) of ozone dissolved in water during ozonation?

R:The concentration (3 g/h) is the ozone production of the ozonizer.

4. Page 6, line 13, the author directly proposed the best choice was 65 μm PDMS / DVB instead of 85 μm CAR/PDMS, 85 μm PA, and 100 μm PDMS. Could the authors provide some data which can explain that 65 μm PDMS/DVB is the best choice?

R: The detail data is shown in Fig 2, which is newly added.

5. Page 6, line 48, the authors mentioned: "Addition of salt was used to compensate on sample recovery". Please give more explanation of why adding salt can increase recovery.

R: The explanation has been given, which is marked green in page 8 line 6,7.

6. Page 12, in the Fig.5, the peaks of DMA and FDMH are not separated, is this related to the temperature rise procedure of GC/MS?

R: I have tried to separate the peaks of DMA and FDMH by rising the temperature of GC/MS. The result was not ideal. I think the reason may be that , the boiling point of DMA is so low, resulting in wider peak and difficult to separate. It may need to be further studied.

7. Page 12, line 60, was the radiolysis experiments done by yourself or by reference? Please indicate if it comes from reference.

R: The radiolysis experiments comes from reference. The reference was added in page 14 line3.

8. Page 13, line 6, the authors mentioned "These experiments results confirm that NDMA formation is due to reactions associated with molecular ozone. Our results

are inconsistent with these results.” I think this statement is confused.

R: The correct expression is “consistent with”, which has been corrected and marked in green in page 14 line 7.

Small comments:

1. There should be a space between the number and the unit.
2. Introduction and conclusion parts of the text should be aligned at both ends.
3. The author should carefully check the content and formatting problems.
4. Page 1, line 19, “N-nitrosodimethylamine” should be “N-Nitrosodimethylamine”.

R: All of them have been corrected.

Appendix B

Dear reviewer,

I am appreciate for your comments and suggestions. The manuscript has been revised based on your comments. The responses of your questions were as follows:

1. Throughout the whole manuscript one space should be given between number and units, e.g.

Page 1, line 33, 65µm should be 65 µm; Page 2, line 14, 0.02mg/L[1] should be 0.02 mg/L [1].

There should one space before the brackets.

R: The brackets have been added, which are marked red.

2. Line 28-32, Formic acid(HCOOH), Methanol(CH₃OH), Acetic acid(CH₃COOH), Nitromethane (NM), N,N-dimethylformamide (DMF), 1,1,5-trimethylformazane (TMFN), Acetaldehyde dimethylhydrazone (ADMH), Formaldehyde (HCHO).

The compound names should be in lower case letters.

R: They are all corrected in lower case letters, which are marked red.

3. Page 4, line 52, Modern samples were samples were... Authors should define what they mean by Modern sample? Is it fresh sample, if so use fresh sample...

R: Yes, It is fresh sample. It has been corrected.

4. Page 5, line 30, before “Results and Discussion” a new sub-section “Statistical analysis” should be include to describe the statistical treatment of data.

R: The part has been added, which is marked red in page 6.

5. Page 6, line 7, The modern sample contains....???

R: I have rewrite this sentence, which marked red in page 7 line 3,4.

6. Page 6, line 44, There were reports...give [Ref. -- ??]

R: The references have been given. (Which is marked red in page 8 line 3).

7. Page 7, Fig. 1, x-axis label should be corrected as Salt concentration (w/v, %)

R: It has been corrected.

8. Page 10, line 46, 9.340mg/L, 3.292mg/L, 5.010mg/L and 7.084mg/L...MDL data should be given in two significant figures only, e.g. 9.3 mg/L, 3.3 mg/L, 5.0 mg/L and 7.1 mg/L...

R: They are all corrected, which are marked red in page 12.

9. Page 11, Table 3, column 5, MDL should be given in two significant figures only. The data in column 2-4 should be given in three significant figures only.

R :They are all corrected, which are marked red in Table 3.

10. Page 17, line 27, before Conclusion, the advantages and disadvantages of the proposed method should be given in comparison to the literature reported methods.

R: The advantages and disadvantages have been discussed in page 18, which are newly added marked red.